# A cross-sectional survey of reproductive, gynecological, and breast health histories and status among people in a provincial prison for women in British Columbia

Clare Heggie[1]*, Martha Paynter[2], Anja McLeod[2], Jessica Liauw[3], Rosann Edwards[2], Fiona Kouyoumdjian[4]

1 Department of Interdisciplinary Studies, University of New Brunswick, Fredericton, New Brunswick, Canada, 2 Faculty of Nursing, University of New Brunswick, Fredericton, New Brunswick, Canada, 3 Department of Obstetrics and Gynaecology, University of British Columbia, Vancouver, British Columbia, Canada, 4 Department of Family Medicine, McMaster University, Hamilton, Ontario, Canada

* clare.heggie@unb.ca

## Abstract

### Objectives

Women are a growing prison population in Canada, yet there is a lack of systematic data collection on the reproductive health of people incarcerated in prisons for women. The objective of this study was to describe a wide range of reproductive, gynecological, and breast health histories, including access to preventive services, among people incarcerated in a provincial prison for women in British Columbia.

### Study design

A cross-sectional survey design was used to meet study objectives.

### Methods

We adapted a survey instrument previously administered in four provincial prisons in Atlantic Canada. The survey consisted of 54 questions about demographics and reproductive, gynecological, and breast health. The survey was administered on paper, in person, and we analyzed data using descriptive statistics.

### Results

Of 75 participants, 48% identified as Indigenous, with a median age of 36 years. Eighty-five percent of participants had ever been pregnant, 72% reported having had an unintended pregnancy and 51% had ever had an abortion. The most used types of contraception included the male condom and birth control pill. Among participants eligible for cervical cancer screening, 48% had a Pap test within the last 3 years. Of

**Data availability statement:** Data cannot be shared publicly because of data containing potentially identifying or sensitive participant information. This restriction on data access is required by the University of New Brunswick Research Ethics Board. Data are available from the UNB Institutional Data Access / Ethics Committee for researchers who meet the criteria for access to confidential data. The Ethics Committee can be contacted at ethics@unb.ca.

**Funding:** Author MP was supported by a University of New Brunswick Harrison McCain Young Scholars Award. This study was also supported by funding received from Women and Gender Equality Canada.

**Competing interests:** The authors have declared that no competing interests exist.

those eligible for screening based on age and provincial guidelines, 40% had ever had a mammogram.

## Conclusions

Findings from this cross-sectional survey highlight health disparities when compared with people in the general community, underscoring the need for routine and systematic data collection on reproductive, gynecological and breast health history and outcomes in this population and the need for collaborative approaches to ensure incarcerated women have access to appropriate and recommended preventive healthcare services.

## Introduction

Women are a growing population in both federal and provincial prisons in Canada [1,2]. Despite this growth, there is a lack of systematic data collection on the sexual and reproductive health outcomes of people incarcerated in prisons for women. The United Nations Rules for the Treatment of Women Prisoners and Non-custodial Measures for Women Offenders, or the Bangkok Rules, require collecting a comprehensive reproductive health history at intake. This history may include current or recent pregnancies, the presence of sexually transmitted infections, and any related reproductive health concerns [3]. The lack of systematic data collection of sexual and reproductive health histories, healthcare access, and outcomes may prevent the planning and delivery of evidence-based and relevant sexual and reproductive healthcare in both prisons and in community settings after release.

Prior global evidence synthesis on the health of people in prisons for women has identified a lack of attention to gynecological and reproductive health outcomes, with research largely focused on substance use, sexually transmitted and blood borne infections, and mental health related outcomes [4–7]. Research is also largely concentrated in the United States, with US-based research identifying variable and limited access to reproductive healthcare for women in prisons, including perinatal healthcare and preventive care such as a cervical and breast cancer screening [8–10]. Prior Canadian research has identified that women in prisons have unmet needs for sexual and reproductive healthcare, though this research has primarily been conducted in Ontario and Alberta. Research conducted using health administrative data identified that women in a provincial prison in Ontario were more likely to be overdue for both cervical cancer and breast cancer screening and less likely to receive adequate antenatal care when compared to women in the general population, [11–13] and a cross-sectional survey of women in a provincial prison in Ontario found that 82% of participants had ever been pregnant, and of these participants, 77% had experienced an unintended pregnancy and 57% had had an abortion [14].

In British Columbia, a survey conducted in 1998 of women incarcerated in a provincial prison similarly found low rates of cervical cancer screening, with almost a third of women reporting not having had a Pap test within the last 2.5 years, despite

75% being willing to undergo Pap testing while incarcerated [15,16]. Evidence from longitudinal community cohorts of women living with HIV and of women sex workers in British Columbia also indicate barriers to accessing both testing and treatment of sexually transmitted and blood borne infections during incarceration [17–19]. Indigenous women are highly over-represented in both federal and provincial prisons in Canada and face additional and heightened barriers to accessing essential reproductive healthcare, including a lack of culturally sensitive care and experiences of institutional and medical racism [20].

Existing Canadian and global evidence demonstrates a need for contemporary Canadian research on under-studied reproductive health histories such as pregnancy and contraception use, gynecological health, and breast health. We aimed to describe a wide range of reproductive, gynecological, and breast health histories among people incarcerated in a provincial prison in British Columbia using a cross-sectional survey. This information could help inform policy, practice, and advocacy to improve the health of women in prison.

## Materials and methods

### Survey instrument

As described elsewhere [21] we had adapted a survey instrument developed by Liauw et al. [14] for a survey of reproductive, gynecological, and breast health outcomes among people in provincial prisons for Atlantic Canada. We modified the adapted survey instrument based on feedback from participants and correctional partners in Atlantic Canada, as well as through discussion with the research team, for the current survey of reproductive, gynecological, and breast health histories and healthcare access among people in a provincial prison for women in British Columbia. Specifically, we added questions about child protection involvement experienced as a child, sexual history and pregnancy intentions prior to custody and expected sexual activity and pregnancy intentions upon release, treatment for sexually transmitted infections, bacterial vaginosis, and yeast infections. We also revised the wording of the survey instrument based on participant feedback on questions that were unclear or confusing. The updated survey comprised 54 questions across 10 sections: Basic Demographic Information; Time in Custody; Parenting; Breastfeeding; Pregnancy; Contraception; Menstrual History; Sexual Health and Sexually Transmitted and Blood Borne Infections; Cervical Screening and Vaccination; and Breast Health. The final section was followed by an open comment box. The full survey instrument is available in Appendix A.

### Setting

In Canada, Correctional Services Canada (CSC) governs federal prisons under the Minister of Public Safety, and each province and territory have their own correctional systems. Provincial and territorial facilities include remand/pretrial custody and provincial sentences of up to two years less a day. A sentence of two years or more results in federal incarceration. There are 7 federal prisons designated for women and 44 provincial and/or territorial facilities designated for women in Canada [22].

The only provincial prison designated exclusively for women in British Columbia, Alouette Correctional Centre for Women, is located in the small city of Maple Ridge, with a population of about 90,000, about a one hour drive from Vancouver. The prison opened in 2004 and was known for its unique Mother Baby Unit program, which ran from 2005–2008 and was officially reopened in 2016. At the time of the survey, there were approximately 75 people incarcerated at the prison, and no babies. The study site, and prior study sites in Atlantic Canada (results from which are reported separately [21]), were selected on the basis of established professional relationships.

### Sample

People who were over the age of 18, able to communicate in English or French, able to provide informed consent, and incarcerated in the provincial prison on the day of survey administration were eligible to participate. Trans and gender diverse people who were incarcerated in these facilities designated for women were eligible to participate.

### Recruitment

We emailed posters detailing the survey topic, length, and honorarium to partners at the study site prior to survey administration. Partners posted the information on the units. Staff at the study site were provided with information about the survey so they could answer questions about participation.

### Data collection procedures

On the day of survey administration, one study team member held a meeting with potential study participants to describe the study and answer any questions. The study team member explicitly stated that the decision whether or not to participate would have no impact on potential study participants' ability to access healthcare and/or other services provided by the institution. After reviewing the study procedures, participants reviewed the consent form and provided verbal consent to participate. The study team member asked participants if they consented to participate and then documented consent by signing a verbal consent form. This process of verbal consent was approved by the Research Ethics Board. The study team member was present while participants completed the survey to clarify survey questions as needed. The survey was administered in a private programming room. Participants were able to skip questions and/or not complete the survey with no impact on receiving an honorarium. Once the survey was completed and returned to the study team member, participants were not able to remove their data. Participants received $20, which was placed on their canteen accounts. Data collection was conducted on November 13th, 2024. This time was selected in discussion with operational staff regarding a time that would be convenient from an operations perspective and to enable participation.

### Analysis

We summarized data using descriptive statistics (i.e., counts and frequencies). Questions with less than five responses were reported as <5 based on the protocol approved by our Research Ethics Board. Qualitative comments written at the end of the survey in the blank text box were transcribed verbatim by the research assistant. We defined participants at risk of unintended pregnancy as participants who reported that they were not trying to conceive and were having vaginal (defined as penis in vagina) intercourse in the three months prior to incarceration or in the six months following release, excluding participants who had undergone hysterectomy. We categorized contraceptive method by typical use effectiveness, where Tier 1 includes intrauterine devices, implant devices, vasectomy and tubal ligation; Tier II includes injectable, oral, ring and patch contraceptives; and Tier III includes barrier contraceptives, spermicide, natural birth control methods, withdrawal, or no method [23]. We did not formally analyze qualitative comments, however, we used this content to contextualize and support key findings.

### Ethical considerations

The survey instrument and study protocol was approved by University of New Brunswick-REB # 2023-124.

### Results

#### Participant demographics

Out of 75 eligible people in custody on the day of survey administration, 68 people completed the survey, for a 91% participation rate. Ineligible people included those who were medically unable to consent and people who may have been attending video court appointments during the day of survey administration. Participants had spent a median of 18 lifetime months in custody and ranged in age from 18–57 years. The median age of participants was 36. The majority of participants identified as women and as heterosexual, with one third identifying as bisexual (Table 1). Almost half identified as Indigenous, with the next most common reported ethnicity being white. Most reported living in an urban area prior to incarceration, and almost half did not expect to have housing or were unsure if they would have housing after release. The vast

**Table 1. Self-reported demographics by participants in a survey in a provincial correctional facility for women in British Columbia, Canada.**

|  | n = 68 |
| --- | --- |
| *Age* | n (% of N = 68) |
| 18-24 | 7 (10.3) |
| 25-29 | 11 (16.2) |
| 30-34 | 11 (16.2) |
| 35-39 | 8 (11.8) |
| 40-44 | 15 (22.0) |
| 45-49 | 8 (11.8) |
| 50-54 | 7 (10.3) |
| No answer provided | <5 |
| *Urban versus rural residence prior to incarceration* |  |
| Urban | 46 (67.6) |
| Rural | 12 (17.6) |
| Unsure/Other | 9 (13.2) |
| No answer provided | <5 |
| *Marital status* |  |
| Married/Common law | 14 (20.6) |
| Single/Never married | 34 (50) |
| Divorced | 9 (13.2) |
| Separated | 7 (10.3) |
| Other | <5 |
| *Average yearly annual income prior to incarceration* |  |
| Less than $20,000 | 38 (55.9) |
| $20,000 - $49,999 | 11 (16.2) |
| Greater than $49,999 | 14 (20.6) |
| Don't know/other/no answer | 5 (7.3) |
| *Gender* |  |
| Woman | 63 (92.6) |
| Non-Binary | <5 |
| Two-Spirit | <5 |
| No answer provided | <5 |
| *Sexual orientation* |  |
| Heterosexual | 40 (58.9) |
| Bisexual | 20 (29.4) |
| Lesbian/Gay | <5 |
| Other/no answer provided | <5 |
| *Racial and Indigenous identity* |  |
| Indigenous | 33 (48.5) |
| Black | <5 |
| Asian | <5 |
| White | 30 (44.1) |
| Other | <5 |
| *Housing status prior to incarceration* |  |
| Had housing | 38 (55.9) |
| Did not have housing | 23 (33.8) |
| Unsure/ no answer provided | 7 (10.3) |

*(Continued)*

**Table 1.** (Continued)

|  | n = 68 |
|---|---|
| *Anticipated housing status upon release* |  |
| Expects to have housing | 34 (50.0) |
| Does not expect to have housing | 12 (17.6) |
| Unsure | 21 (30.8) |
| No answer provided | <5 |
| *Total lifetime months spent in custody* |  |
| 0-6 | 21 (30.9) |
| 7-12 | 14 (20.6) |
| 13-18 | <5 |
| 19-24 | <5 |
| Greater than 24 | 20 (29.4) |
| No answer provided | 5 (7.3) |

majority of participants lived on an average annual income of $50,000 Canadian dollars or less. See Table 1 for participant demographics.

## Parenting and children

Sixty-six percent (n = 45) of participants reported having children. Participants reported having a total of 100 children across the entire sample, with a median of one child per participant. Eighteen percent of participants (n = 12) were the primary caregiver for their children under 18 prior to incarceration. Among participants with children, 53% (n = 24) reported having any child protection involvement as a parent prior to incarceration. See Table 2.

## Pregnancy

Eighty-five percent of participants had ever been pregnant (n = 58). Participants had a median lifetime number of 3 pregnancies, 72% (n = 49) of participants had ever had an unintended pregnancy, and 51% (n = 35) of participants had ever had an abortion. The median number of lifetime abortions per person was 1.5. See Table 3.

## Contraception

Most participants reported having ever used methods of contraception that are generally considered tier II or III (i.e., medium or low) effectiveness. [23] The most common methods ever used were oral contraceptives (n = 45), male condom (n = 39), and withdrawal (n = 39). In the three months prior to current incarceration, the most common methods were hormonal intrauterine device (n = 11), withdrawal (n = 11), and male condom (n = 10). Thirty-four percent of participants (n = 23) were not using any contraceptive method in the three months prior to current incarceration. See Table 4.

For participants at risk of unintended pregnancy in the three months prior to incarceration (n = 29), 69% (n = 20) reported use of any of the following contraceptive methods in the three months prior to incarceration: barrier contraception, hormonal contraceptives, intrauterine device, or tubal ligation. Thus, 31% (n = 9) of participants at risk for unintended pregnancy were not using reliable contraception in the three months prior to incarceration. For participants at risk of unintended pregnancy in the six months after release (n = 24), 33% (n = 8) did not plan to use or were unsure about using contraception after release, and 66% (n = 16) planned to use any of the following contraceptive methods in the six months after release: barrier contraception, hormonal contraceptives, intrauterine devices, or tubal ligation.

**Table 2. Self-reported children and parenting status by participants in a survey in a provincial correctional facility for women in British Columbia, Canada.**

| Parenting status | n=68 |
| --- | --- |
| Has children under the age of 1 | 0 |
| Has children aged 1–5 | 7 (10.3) |
| Has children aged 6–17 | 21 (30.9) |
| Has children aged 18+ | 23 (33.8) |
| Does not have children | 23 (33.8) |
| Respondent was primary caregiver for children prior to incarceration (children under 18) | n=68 |
| Yes | 12 (17.6) |
| No | 28 (41.2) |
| Other/no answer provided | 6 (8.8) |
| Not applicable | 22 (32.3) |
| Other | 14 (31.1) |
| Any child protection involvement as a parent prior to incarceration among participants with children (children under 18) | n=45 |
| Yes | 24 (53.3) |
| No | 21 (46.7) |
| Expected to have child protection involvement upon release among participants with children (children under 18) | n=45 |
| Yes | 3 (6.7) |
| No | 33 (73.3) |
| Unsure/no answer provided | 9 (20.0) |

**Table 3. Self-reported pregnancy and breastfeeding history by participants in a survey in a provincial correctional facility for women in British Columbia, Canada.**

| Unintended pregnancy history * | n=68 |
| --- | --- |
| Has had an unintended pregnancy | 49 (72.0) |
| Has never had an unintended pregnancy | 16 (23.5) |
| No answer provided | 3 (4.4) |
| Pregnancy outcome history | n=68 |
| Had a live birth/child born alive | 44 (64.7) |
| Had a stillbirth** | 6 (8.8) |
| Had a miscarriage or ectopic*** | 26 (38.2) |
| Had an abortion**** | 35 (51.5) |
| Has ever breastfed | n=68 |
| Yes | 35 (51.5) |
| No | 10 (14.7) |
| Not applicable/no answer provided | 23 (33.8) |
| Has ever pumped milk | |
| Yes | 28 (41.2) |
| No | 17 (25.0) |
| Not applicable/no answer provided | 23 (33.8) |

* Defined as a pregnancy you didn't want, didn't plan for, and/or or happened at the wrong time.

**Defined as a pregnancy that went beyond 20 weeks, but the baby died before being born.

***Defined as a spontaneous loss before 20 weeks.

****Defined as a pregnancy that was ended on purpose.

**Table 4. Self-reported contraception use by participants in a survey in a provincial correctional facility for women in British Columbia, Canada N = 68.**

| Contraceptive Method | Ever used method | Used method in 3 months prior to current admission to custody | In the 3 months prior to current admission to custody by participants at risk of unintended pregnancy* n = 29 | Tier of effectiveness |
|---|---|---|---|---|
| Copper intrauterine device | 11 (16.2) | <5 | 0.0 | I |
| Hormonal intrauterine device | 20 (29.4) | 11 (16.2) | 10 | I |
| Hormonal implant | <5 | 0.0 | 0.0 | I |
| Tubal ligation | 7 (10.3) | 5 (7.3) | <5 | I |
| Hysterectomy | <5 | <5 | N/A | I |
| Vasectomy | <5 | <5 | 0.0 | I |
| Oral contraceptive | 45 (66.2) | 6 (13.2) |  | II |
| Contraceptive patch | 5 | <5 | <5 | II |
| Vaginal ring | <5 | 0.0 | 0.0 | II |
| Injectable contraceptive | 27 (39.7) | 5 (7.4) | <5 | II |
| Emergency contraception | 11 (16.2) | <5 | <5 | II |
| Male condom | 39 (57.4) | 10 (14.7) | 9 (31.0) | III |
| Female condom | 6 (8.8) | <5 | <5 | III |
| Contraceptive sponge | <5 | <5 | 0.0 | III |
| Cervical cap | 0.0 | 0.0 | 0.0 |  |
| Diaphragm | <5 | <5 | 0.0 | III |
| Spermicide | 5 | <5 | <5 | III |
| Natural birth control methods | 6 | <5 | <5 | III |
| Breastfeeding | 5 | <5 | 0.0 | III |
| Withdrawal | 33 (48.5) | 11 (16.2) | 10 (34.5) | III |
| Non-vaginal intercourse | 5 (7.4) | 0.0 | 0.0 | III |
| Abstinence | 11 (16.2) | <5 | <5 | III |
| Other | <5 | <5 | <5 |  |
| None | <5 | 23 (33.8) | 5 (17.2) |  |

* Defined as a pregnancy you didn't want, didn't plan for, and/or happened at the wrong time.

## Gynecological health

Forty-eight percent of participants (n = 33) had ever experienced symptoms of extremely painful periods or extremely heavy bleeding and 29% (n = 20) of all participants had ever discussed any menstrual concerns with a healthcare provider. Sixteen percent (n = 11) felt they did not have adequate access to menstrual products while in custody.

Fifty-one percent (n = 35) of participants reported having an STI test in the past 6 months, 41% (n = 28) reported being offered an STI test upon admission to the institution, and 44% (n = 30) stated they currently wanted an STI test. Forty-three percent (n = 29) had ever had a positive test for chlamydia, 19% (n = 13) for gonorrhea, 18% (n = 12) for syphilis, and 25% (n = 17) for hepatitis C. There were less than 5 people who had ever received a positive HIV test. Fifty-seven percent of participants (n = 39) had ever been diagnosed with a yeast infection and 41% (n = 28) had ever been diagnosed with bacterial vaginosis. Twenty-eight percent (n = 19) of all participants and 30% (n = 16) of participants aged 45 or younger had ever received the HPV vaccine. Of 60 people who would have been eligible for cervical cancer screening based on their age and provincial guidelines, [24] 48% (n = 29) had had a Pap test within the last 3 years. Twenty-seven percent (n = 18) of participants had ever had an abnormal pap test result (Table 5).

**Table 5. Self-reported gynecological history by participants in a survey in a provincial correctional facility for women in British Columbia, Canada, N = 68.**

| Ever experienced symptoms of dysmenorrhea | N (%) |
|---|---|
| Yes | 33 (48.5) |
| No | 34 (50.0) |
| No answer provided | <5 |
| Has adequate access to menstrual products while in custody | |
| Yes | 47 (69.1) |
| No | 11 (16.2) |
| Unsure/ no answer provided | 10 (14.7) |
| STBBI testing and treatment history | |
| Has had a positive test for chlamydia | 29 (42.6) |
| Received treatment for chlamydia | 23 (33.8) |
| Has had a positive test for gonorrhea | 13 (19.1) |
| Received treatment for gonorrhea | 9 (13.2) |
| Has had a positive test for syphilis | 12 (17.6) |
| Received treatment for syphilis | 10 (14.7) |
| Has had a positive test for HIV | <5 |
| Received treatment for HIV | <5 |
| Has had a positive test for hepatitis C | 17 (25.0) |
| Received treatment for hepatitis C | 11 (16.2) |
| Had positive test for BV | |
| Yes | 28 (41.2) |
| No | 30 (44.1) |
| Unsure/no answer provided | 10 (14.7) |
| Had positive test for yeast infection | |
| Yes | 39 (57.4) |
| No | 20 (29.4) |
| Unsure/no answer provided | 9 (13.4) |
| Received any HPV vaccination | |
| Yes | 19 (27.9) |
| No | 38 (55.9) |
| Unsure | 9 (13.2) |
| No answer provided | <5 |
| Has had a Pap test within the last three years | |
| Yes | 32 (47.1) |
| No | 9 (13.2) |
| Unsure | 20 (29.4) |
| No answer provided | 7 (10.3) |
| Has ever had an abnormal Pap test result | |
| Yes | 18 (26.5) |
| No | 25 (36.8) |
| Unsure | 21 (30.9) |
| No answer provided | <5 |

## Breast health

Eighteen percent of participants (n = 12) had ever noticed breast abnormalities. Of the 30 people who would have been eligible for screening based on their age and local guideline, [25] 40% (n = 12) had ever had a mammogram.

## Discussion

This was the first cross-sectional survey of reproductive, gynecological and breast health histories, including access to preventive services, in a provincial prison for women in British Columbia, and the third cross-sectional survey of reproductive health histories and outcomes in provincial prisons in Canada. Consistent with prior surveys conducted in Ontario and in Atlantic Canada, we identified high lifetime rates of unintended pregnancy and abortion [14,21]. Unsurprisingly, we also found that people in prisons for women experience intersecting structural determinants of health inequity, including housing precarity and low income, which may impede access to care. Previous research synthesis regarding sexual and reproductive health among incarcerated women in Canada have identified that research with this population is dominated by a focus on HIV and sexually transmitted and blood borne infections, with little attention to other reproductive health outcomes [26]. The results of this survey highlight the importance of attention to other health needs such as pregnancy care, family planning, cervical screening, and menstrual equity.

In comparing results to available regional and national statistics in the general population, we identified several key differences in demographics and health history. For example, over half of all participants had ever had an abortion, compared to national estimates that 1 in 3 women in Canada will have an abortion in her lifetime, [27] and 72% of participants had ever had an unintended pregnancy, compared to national estimates that 40% of pregnancies are unintended [28]. While we did not ask about HPV vaccine completion, 28% of participants reported ever having any HPV vaccine, compared to a national HPV vaccination completion rate of 64% [29]. Only 51% of eligible participants in this study reported having a pap test within the last three years, compared with approximately 60–75% of eligible people receiving screening in the general BC and Canadian populations [30,31]. We also identified substantial over-representation of people who are Indigenous and people who identify as bisexual or lesbian when compared to national general population estimates [32,33].

The results of this survey add to existing evidence from other Canadian provinces, suggesting disparities in sexual and reproductive health status between people who are incarcerated in provincial prisons in Canada and the general population. In particular, our findings call for action to address barriers to accessing sexual and reproductive health services both while incarcerated and in the community before and after incarceration, such as a cervical cancer screening, HPV vaccination, and contraception. The overrepresentation of people who are Indigenous and people who identify as bisexual or lesbian demonstrates the need for the planning and delivery of healthcare programming and services that are sensitive to cultural needs.

This study has several limitations. Although this small sample may not be representative of people incarcerated in prisons for women in BC or across other provinces and territories, data on women in custody with BC Corrections between 2013–2023 reported that 45% of women in custody identified as Indigenous, and the majority were between the ages of 30 and 34, suggesting that our sample may be somewhat representative demographically [34]. Further, participants are not representative of people currently incarcerated in federal institutions, in other custodial settings such as immigration detention centres, or on community supervision. Due to the small sample size, we were not able to compare survey responses between Indigenous and non-Indigenous participants. Although a study team member was present during survey administration to answer questions and we made all efforts to use plain language, the use of some clinical terms was unavoidable and may have been confusing. This study is also limited by recall bias, as surveys asked questions about historical access of healthcare services and lifetime experiences of pregnancy, abortion and contraception use.

To address some of these limitations, we recommend routine and systematic data collection on reproductive, gynecological and breast health history and outcomes in this population across multiple jurisdictions. This data would allow for better understanding of reproductive health histories and status for services amongst provincially incarcerated women broadly and would help to identify region-specific needs and opportunities for the development of healthcare programming and policy. Future research could also seek the perspectives of prison healthcare staff and leadership to better understand the current availability and provision of services, and to identify resource and knowledge needs with respect to reproductive, gynecological and breast health.

## Conclusions

This cross-sectional survey conducted in a provincial prison for women in British Columbia builds on our prior survey in Atlantic Canada to present a clearer understanding of the reproductive, gynecological and breast health histories and status among people incarcerated in prisons for women in Canada. We identified high rates of reproductive health needs among women in prison, suggesting a need for essential healthcare services including family planning care, cervical cancer screening, mammograms, and routine STI testing. Our findings underscore the need for routine and systematic data collection on reproductive, gynecological and breast health history and outcomes in this population. Future research and healthcare planning must be attentive to the specific healthcare delivery needs of Indigenous and 2SLGBTQ+ people due to their substantial over-representation in the provincial prison system.

## Acknowledgments

The authors would like to thank Margaret Erickson and Lin Tong for their contributions to this study.

## Author contributions

**Conceptualization:** Clare Heggie, Martha Paynter, Anja McLeod, Jessica Liauw, Rosann Edwards, Fiona Kouyoumdjian.

**Formal analysis:** Clare Heggie, Martha Paynter, Anja McLeod.

**Investigation:** Martha Paynter.

**Methodology:** Martha Paynter, Jessica Liauw, Fiona Kouyoumdjian.

**Project administration:** Anja McLeod.

**Resources:** Martha Paynter.

**Supervision:** Martha Paynter, Jessica Liauw, Fiona Kouyoumdjian.

**Writing – original draft:** Clare Heggie.

**Writing – review & editing:** Martha Paynter, Anja McLeod, Jessica Liauw, Rosann Edwards, Fiona Kouyoumdjian.

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
