## [Decision Letter · Decision Letter 0]

15 Sep 2025

Dear Dr. Heggie,

We look forward to receiving your revised manuscript.

Kind regards,

Andrea K. Knittel

Academic Editor

PLOS ONE

Journal Requirements:

2. Please provide additional information regarding the considerations  made for the prisoners included in this study. For instance, please discuss whether participants were able to opt out of the study and whether individuals who did not participate receive the same treatment offered to participants.

3. In the ethics statement in the Methods, you have specified that verbal consent was obtained. Please provide additional details regarding how this consent was documented and witnessed, and state whether this was approved by the IRB

**Additional Editor Comments:**

Please attend to each issue raised by the reviewers in your re-submission.

Reviewers' comments:

Reviewer's Responses to Questions

**Comments to the Author**

1. Is the manuscript technically sound, and do the data support the conclusions?

Reviewer #1: Yes

Reviewer #2: Yes

2. Has the statistical analysis been performed appropriately and rigorously?

Reviewer #1: Yes

Reviewer #2: Yes

3. Have the authors made all data underlying the findings in their manuscript fully available?

Reviewer #1: Yes

Reviewer #2: Yes

4. Is the manuscript presented in an intelligible fashion and written in standard English?

Reviewer #1: Yes

Reviewer #2: Yes

Reviewer #1: Nice work that adds to the literature about health disparities in a very vulnerable population. Thank you for your willingness to do this work.

Not reported was whether there were any differences in the past histories of Indigenous and other women. This would have implications that are beyond incarcerated populations.

A small point: as a limitation, you say, “Participants may not be representative of the population of people incarcerated 281 across all prisons in British Columbia, or across different provinces and territories.” Do you not have access to demographics of incarcerated women in the province?

Two comments about language. This is not a survey about health outcomes. It is about sexual/reproductive health histories and use of/access to preventive services. There are no outcomes here.

Also, I would strongly suggest that the results do much more than “underscore the need for routine and systematic data collection” as written in your Abstract. They underscore the need for the health and corrections systems work together to assure that incarcerated women have access to appropriate and recommended preventive services! What else is the point of the research data collection???

Please strengthen the language of your conclusion.

Reviewer #2: Thank you for this pilot survey study of people experiencing incarceration in Canada and their access to services. This study is extremely important and highlights papers from the United States on challenges to accessing GYN healthcare services in the carceral setting.

1. Introduction- There are several papers on GYN outcomes in carceral settings in the US. Specifically including those and their highlighting of social/cultural/historic barriers worsening outcomes for specific vulnerable populations as a part of the introduction would highlight the parallels you note for indigenous and gender diverse folks. There are also several papers on indigenous health outcomes that offer reasons for that variation specifically in Canada. I think the introduction can be expanded to explain why you created such a broad survey. Some information about the carceral system for women in Canada would be helpful to put this specific provincial prison in context. Some grammar change for flow would be helpful.

2. Methods- Very clear and direct. Would you be able to share your survey instrument?

3. Results/Conclusion--I would remove the "NO: categories from your questions. Just list the "YES" and "don't know" so that we can reduce the size of tables and streamline appearance. Consider not reporting data <5. Pap tests can happen every 5 years--why does q3 years raise concern? Given the majority of indigenous identifying folks, is it possible to break some data down to specifically look at indigenous versus general population responses to see if there is a difference in experience? Do their qualitative responses add any details explaining their experience? I only ask because one of the major conclusions of this paper is the need for culturally sensitive care. I'm not sure if its lack of basic access to resources/racism or culturally concordant care that's driving these outcomes and I'm also not convinced we can make that conclusion with the current data. I think a paragraph on recommended next steps to address the limitations of this paper may be useful. Is there a way to get data on other provincial prisons in Canada to generally discuss reproductive access and its challenges? Do all prisons have access to a gynecologist? Is there any way to get this information? It may be that this data does not exist, but perhaps highlighting what next steps would be needed (again) would help clarify some of the statements in your conclusion. Please include recall bias in your limitations.

Thank you for your incredible work! I appreciate your time with this review.

**Do you want your identity to be public for this peer review?** For information about this choice, including consent withdrawal, please see our Privacy Policy

Reviewer #1: No

Reviewer #2: No

---

## [Author Response · Author response to Decision Letter 1]

27 Oct 2025

Response to reviewers attached in revision files

---

## [Editor Report · Decision Letter 1]

12 Nov 2025

Dear Dr. Heggie,

We look forward to receiving your revised manuscript.

Kind regards,

Andrea K. Knittel

Academic Editor

PLOS ONE

Journal Requirements:

Additional Editor Comments :

Thank you for your thought and care in responding to the reviewers comments. I identified a few small instances where the language should be further changed to be concordant with the other recommendations. I am using the line numbers from the tracked changes version of the revised document:

Lines 119-121: Still says "health outcomes." Please revise for consistency.

Line 291-308: It is confusing to say that your results are not representative and then state that they are! I would recommend removing the clause on lines 291-292 that reads "While this small sample s not representative of all people in provincial prisons in Canada," and just start with "The results of this survey..." Then you can condense the discussion of representation into the limitations paragraph. For that paragraph, I think it would be clearer to say something like "Although this small sample may not be representative of all people housed in women's facilities across Canada, there are some reassuring data that support representativeness within the province." And then talk about the demographic data you have for BC and the high response rate. Then the limitations about other provinces, the federal facilities, and community supervision.

Lines 301-315: Please add to the limitations some version of the text in the response to reviewers document that explains that the sample size was too small to draw comparisons between Indigenous and non-Indigenous Canadians in your sample.

---

## [Author Response · Author response to Decision Letter 2]

18 Nov 2025

Thank you for your review of this paper. We have responded to the additional reviewer comments in the attached response to reviewers document.

---

## [Editor Report · Decision Letter 2]

10 Dec 2025

A cross-sectional survey of reproductive, gynecological, and breast health histories and status among people in a provincial prison for women in British Columbia

PONE-D-25-38265R2

Dear Dr. Heggie,

We’re pleased to inform you that your manuscript has been judged scientifically suitable for publication and will be formally accepted for publication once it meets all outstanding technical requirements.

Kind regards,

Andrea K. Knittel

Academic Editor

PLOS One

Additional Editor Comments (optional):

Thank you for your careful attention to all of the reviewer and editor comments.
---

## [Editor Report · Acceptance letter]

PONE-D-25-38265R2

PLOS One

Dear Dr. Heggie,

I'm pleased to inform you that your manuscript has been deemed suitable for publication in PLOS One. Congratulations! Your manuscript is now being handed over to our production team.

Kind regards,

on behalf of

Dr. Andrea K. Knittel

Academic Editor

PLOS One